# Distributionally Robust Graph-based Recommendation System

## ABSTRACT

With the capacity to capture high-order collaborative signals, Graph Neural Networks (GNNs) have emerged as powerful methods in Recommender Systems (RS). However, their efficacy often hinges on the assumption that training and testing data share the same distribution (*a.k.a.* IID assumption), and exhibits significant declines under distribution shifts. Distribution shifts commonly arises in RS, often attributed to the dynamic nature of user preferences or ubiquitous biases during data collection in RS. Despite its significance, researches on GNN-based recommendation against distribution shift are still sparse.

To bridge this gap, we propose DR-GNN that incorporates Distributional Robust Optimization (DRO) into the GNN-based recommendation. DR-GNN addresses two core challenges: 1) To enable DRO to cater to graph data intertwined with GNN, we reinterpret GNN as a graph smoothing regularizer, thereby facilitating the nuanced application of DRO; 2) Given the typically sparse nature of recommendation data, which might impede robust optimization, we introduce slight perturbations in the training distribution to expand its support. Notably, while DR-GNN involves complex optimization, it can be implemented easily and efficiently. Our extensive experiments validate the effectiveness of DR-GNN against three typical distribution shifts.

## KEYWORDS

Graph Recommendation, Out of distribution, Robust

**ACM Reference Format:**
Anonymous Author(s). 2018. Distributionally Robust Graph-based Recommendation System. In *Proceedings of Make sure to enter the correct conference title from your rights confirmation emai (Conference acronym 'XX)*. ACM, New York, NY, USA, 11 pages. https://doi.org/XXXXXXX.XXXXXXX

## 1 INTRODUCTION

In recent years, Graph Neural Networks (GNNs) have attracted considerable attention in field of recommendation systems (RS) [1, 4, 7, 10, 20, 49]. GNN-based recommendation methods often follow this typical pipeline: 1) constructing a graph based on users' historical interactions in training data; 2) utilizing multi-layered GNNs on the constructed graph to derive user and item embeddings; (3) generating recommendations based on the embedding similarities. Owing to their capability to capture high-order collaborative signals, GNN-based methods have achieved state-of-the-art performance in collaborative recommendation.

However, a pervasive assumption underpinning many GNN-based methods is that testing data holds the same distribution as training data (*a.k.a.* IID assumption). Unfortunately, this assumption often fails in practical, as distribution shifts are ubiquitous in real-world scenarios[13, 14, 39]. Such shifts can arise from: 1) the evolving nature of user preferences[36] — *e.g.,* users may increasingly favor luxury goods as their income increases, sidelining economical alternatives; 2) intrinsic biases within RS[5] — *e.g.,* popular items are often over-represented in the training data due to their elevated visibility. These distributional shifts compromise the effectiveness of the constructed graph, thereby undermining the efficacy of recommendations. As illustrated in Figure 1, although GNN applications demonstrate significant advancements under IID testing scenarios (40.12% and 6.97% improvements across two datasets), these gains drop dramatically when faced with distribution shift (40.12% → 13.57% and 6.97% → 2.76%). This inspires an important question: *Can GNN-based recommendation methods be refined to better manage distribution shift?*

Despite urgency, existing literature on this problem remains sparse. The majority of relevant research can be broadly classified into two categories:

- **Recommendation Methods against distribution shift**: Some recent efforts have employed invariant learning[39, 47, 50] or causal inference [13, 36] to boost the model robustness against distribution shift. Nonetheless, these methods are not tailored for GNN-based methods. As a result, distributional biases remain entrenched in the constructed graphs, distorting the learned embeddings and recommendations accordingly.
- **Robust GNN-based Recommendation Methods:** Some researchers focus on the robustness of GNN-based recommendation methods, employing graph augmentation[3, 41, 43], graph reconstruction[8], or edge weight adjustments[52]. Nevertheless, these methods primarily address interaction noise or popularity bias, rather than universal distribution shifts. Furthermore, most of these methods are heuristic, *e.g.,* leaning on manual-designed augmentations[8] or edge weights[52]. Such reliance on human intuition not only lacks solid theoretical guarantees but also hinders their real-world applicability, especially when distribution shifts are multifaceted and unpredictable. This leaves practitioners in a quagmire, forced to experimentally tweak strategies without clear guidance.

Given these limitations, there is an imperative need for the design of theoretically-grounded GNN-based methods tailored for against distribution shifts. Inspired by the efficacy of *Distributionally Robust Optimization* (DRO), we are inclined to incorporate DRO into this task. DRO offers a theoretical framework that optimizes a model across not just the observable training distribution, but also across a broader family of distributions, effectively accounting for distribution shift. The pipeline of DRO can be delineated as: 1) identifying the hardest distribution over a given distribution family

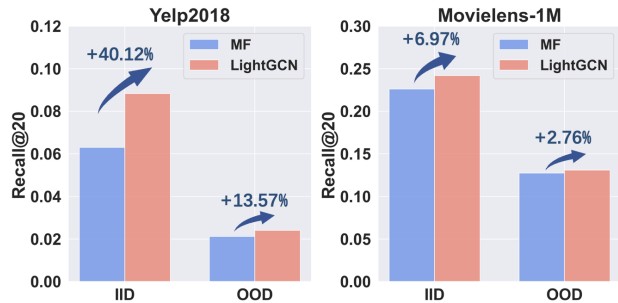

**Figure 1: The performance comparison of MF [17] and LightGCN [10] under both in-distribution (IID) and out-of-distribution (OOD) testing scenarios. For the OOD testing, here we introduce popularity shift in Yelp2018 and temporal shift in Movielens-1M. More details about experimental setting refer to section 4.**

according to the current optimization context; and 2) optimizing the model based on this identified distribution. However, transferring this appealing technique to GNN-based recommendation poses two primary challenges:

**Challenge 1**: Current DRO methods are mainly applied in data with Euclidean structures (*e.g.,* images[23, 25, 34, 35], sentences[21]). Adapting them to manage graph-structured data remains an an open problem. More critically, the effect of graph structure is tangled with the complex graph neural networks, complicating the derivation of the hardest distribution within DRO.

**Challenge 2**: Another challenge arises from the typically sparse nature of recommendation data[12]. With nodes often having limited neighbors, the feasible distribution family for DRO is inherently constrained, which might dampen its robustness and overall performance[28].

To bridge these gaps, we propose DR-GNN, a novel method that seamlessly integrates DRO into GNN-based recommendation. To address the first challenge, we harness insights from graph filtering theories[18], recasting the GNN into an equivalent graph smoothing regularizer that penalizes the distance between adjacent nodes' embeddings. Through this perspective, we incorporate DRO into this regularizer, enhancing the GNN's robustness against shifts existing in neighbor distributions. For the second challenge, we propose a strategy to augment the observed neighbor distribution with slight perturbations. Notably, while DR-GNN does introduce intricate nested optimization, meticulous simplification ensures its implementation remains easy and efficient — primarily drawing from similar nodes as new neighbors and adjusting edge weights according to embedding distances. Our rigorous theoretical analysis proves that if the divergence between training and testing distributions is bounded, the robustness of DR-GNN can be guaranteed.

Our contributions are summarized as follows:

- Exploring the less-explored task of GNN-based recommendation with distribution shift, and revealing the limitations and inadequacies of existing methods.

- Proposing a new GNN-based method DR-GNN for OOD recommendation, which seamlessly integrate DRO in GNN-based methods.
- Conducting extensive experiments to validate the superiority of the proposed method against three types of distribution shift (*i.e.,* popularity shift, temporal shift and exposure shift).

## 2 PRELIMINARY

In this section, we present the background of Graph-based Recommender System and Distributionally Robust Optimization.

### 2.1 GNN-based Recommender Systems

Given a data of user-item interactions $\mathcal{D} = \{u, i, r_{u,i} \mid u \in \mathcal{U}, i \in \mathcal{I}\}$, where $\mathcal{U}$ denotes the set of users, $\mathcal{I}$ denotes the set of items, and $r_{u,i} = \{0, 1\}$ indicates whether user $u$ has interacted with item $i$. We can represent user-item interaction data in the form of a bipartite graph $\mathcal{G} = (\mathcal{V}, \mathcal{E})$ where $\mathcal{V} = \mathcal{U} \cup \mathcal{I}$ denotes the user/item nodes and $\mathcal{E}$ denotes the edge set representing the interactions between users and items. Let $A \in \mathbb{R}^{(|\mathcal{U}|+|\mathcal{I}|) \times (|\mathcal{U}|+|\mathcal{I}|)}$ denote the adjacency matrix of graph $\mathcal{G}$, where $A_{u,i} = 1$ if $r_{ui} = 1$, and $A_{u,i} = 0$ otherwise. Let $\mathcal{N}(u) = \{i \in \mathcal{I} | A_{u,i} = 1\}$ represents the item set that the user $u$ has interacted with. We also define $P_u$ as the distribution of the neighbors of $u$, a uniform distribution over neighbors. Let $d_u$ represent the degree of user $u$. The goal of GNN-based RS is to learn high-quality embeddings from the graph $\mathcal{G}$ and accordingly make accurate recommendations.

**LightGCN.** As a representative GNN-based recommendation methods, LightGCN learns user/item representations following the general message passing of GNNs. Nevertheless, it removes feature transformation and non-linear activations, as they tend to increase overfitting risks without enhancing performance. let denote the embeddings of users and items as $E \in \mathbb{R}^{(|\mathcal{U}|+|\mathcal{I}|) \times c}$ ( $c$ is the dimension of representations). In LightGCN, the final embeddings were obtained via $K$-th propagation layers, which can be abstracted as:

$$E^{(k)} = \widetilde{A} E^{(k-1)}, \tag{1}$$

where $E^{(k)}$ denotes the embeddings after $k$-th propagation layers and $\widetilde{A} = \left(D^{-\frac{1}{2}} A D^{-\frac{1}{2}}\right)$ denotes as the normalized adjacency matrix. $D$ is the diagonal node degree matrix. The notation L denote the Laplacian matrix of graph $\mathcal{G}$, *i.e.,* $L = I - \widetilde{A}$.

To facilitate understanding, considering the prominence and widespread application of LightGCN, we simply take it as the backbone for analysis.

### 2.2 Distributionally Robust Optimization

The machine learning model's success is based on the IID assumption, *i.e.,* training data and testing data are drawn from the same distribution. However, the assumption fails to hold in many real-world applications, leading a sharp drop in performance. Distributionally Robust Optimization (DRO) addresses the issue by considering the uncertainty of the distribution. Specifically, instead of focusing on performance under a single observed data distribution, DRO aims to ensure that the model performs well across a range of potential data distributions. It first identifies the worst distribution(*i.e.,*

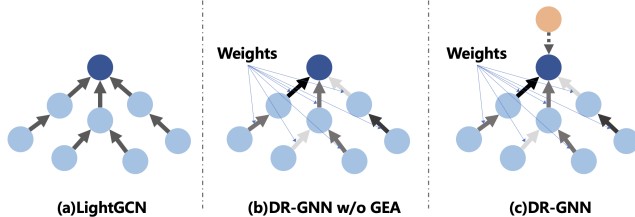

(a)LightGCN  (b)DR-GNN w/o GEA  (c)DR-GNN

**Figure 2: Illustration of how DR-GNN augments LightGCN: it gives edge weights during graph aggregation and introduces new nodes as neighbors.**

the loss function is maximized under the expectation of the worst-case distribution) in the family of distributions within a predefined range around the empirical data distribution, and then optimizes the objective function under that distribution, thereby ensuring the robustness of the model under unfavorable conditions. Formally, DRO is solved in a bi-level optimization via playing the min-max game on the objective function $\mathcal{L}(x; \theta)$, in which $x$ represents the data variable and $\theta$ represents the model parameter to be optimized. The optimization objective of DRO is as follows:

$$\hat{\theta} = \underset{\theta}{\arg\min} \left\{ \max_{P \in \mathbb{P}} \mathbb{E}_{x \sim P}[\mathcal{L}(x; \theta)] \right\}$$

$$\mathbb{P} = \{P \in \mathbb{D} : D(P, P_o) \le \eta\} \tag{2}$$

where $\mathbb{D}$ denotes the set of all distributions, $\eta$ denotes robust radius. The *uncertainty set* $\mathbb{P}$ is composed of all distributions within the robust radius distance from $P_o$, and DRO tried to identify *the worst-case distribution* $P^*$ from a family of eligible distributions $\mathbb{P}$ by maximization. The function $D(., .)$ measures the distance between two distributions *e.g.,* KL-divergence.

## 3 METHODS

In this section, we detail the proposed DR-GNN (subsection 3.1&3.2), followed by the theoretical analyses to demonstrate its robustness (subsection 3.3). Finally, we discuss the connections our DR-GNN with existing methods (subsection 3.4). The schematic diagram of DR-GNN is depicted in Figure 4.

### 3.1 Distributionally Robust GNN

It is highly challenging to directly apply DRO in GNN-based recommendation methods, as the graph structure is tangled with the complex GNN procedure. To tackle this problem, we draw inspiration from graph filtering theories[18] and recast the GNN into an equivalent graph smoothing regularizer.

**LightGCN as a Graph Smoothness Regularizer.** By analyzing LightGCN from the graph signal filtering perspective, we have the following important lemma:

LEMMA 1. *Performing graph aggregation in LightGCN is equivalent to optimizing the following graph smoothness regularizer using*

*gradient descent with appropriate learning rate:*

$$\mathcal{L}_{smooth} = \frac{1}{2} \sum_u \mathbb{E}_{v \sim P_u} [d_u g(u, v; \theta)] \tag{3}$$

$$g(u, v; \theta) = \left\| \frac{\mathrm{E}_u}{\sqrt{d_u}} - \frac{\mathrm{E}_v}{\sqrt{d_v}} \right\|_F^2 \tag{4}$$

Here, $g(u, v; \theta)$ signifies the Fibonacci norm between the normalized embeddings, with $\theta$ representing the model parameters. $P_u$ denotes the distribution of the neighbor nodes of $u$.

The proof can be found in Appendix A.1. This lemma clearly elucidates the effect of the graph neural network — graph aggregation tend to draw the embeddings of neighbors closer. Moreover, the impact of distribution shifts on GNNs are highlighted. Taking the popularity shift (*a.k.a.* popularity bias) as an example, user representations may become excessively aligned with popular items, thereby exacerbating the Matthew effect, as demonstrated in[9]. For the convenience of subsequent analysis, we denote $\mathcal{L}_{smooth}(u) = \mathbb{E}_{v \sim P_u} [d_u g(u, v; \theta)]$ as the smoothness regularizer on the specific node $u$.

**Leveraging DRO in the Regularizer.** Holding the view of GNN as a regularizer, we further introduce DR-GNN, a model that incorporates DRO to enhance its robustness against distribution shifts. Following the definition of DRO Eq.(2), the objective function of the proposed DR-GNN can be formulated as:

$$\min_{\theta} \mathcal{L}_{DRO\_smooth}(u) = \min_{\theta} \max_{P} \mathbb{E}_{v \sim P} [d_u g(u, v; \theta)]$$

$$\text{s.t. } D_{KL}(P, P_u) \le \eta \tag{5}$$

DR-GNN engages in a min-max optimization: 1) it identifies the most difficult distribution over a set of potential distributions. It is defined in the vicinity of the observed neighbor distribution subject to the constraint $D_{KL}(P, P_u) \le \eta$; 2) Subsequently, the model's optimization is performed on this identified hardest distribution instead of original observed distribution. This strategy intrinsically incorporates potential distribution shifts during training, thereby naturally exhibits better robustness against distribution shifts. We will further validate this point in both theoretical analyses (Section 3.3) and empirical experiments (Section 4).

**Efficient Implementation.** Despite the promise, the objective of DR-GNN involves the complex nested optimization, which may incur heavily computational overhead. Fortunately, it can be largely simplified with the following lemma:

LEMMA 2. *The bi-level optimization problem of Eq.(5) can be transformed into optimizing:*

$$\min_{\theta} \mathcal{L}_{DRO\_smooth}(u) = \min_{\theta} \mathbb{E}_{v \sim P_u^*} [d_u g(u, v; \theta)] \tag{6}$$

*where $\alpha$ represents the optimal Lagrange coefficient of the constraint $D_{KL}(Q, P_u) \le \eta$, which can be regarded as a surrogate parameter of $\eta$. The worst-case distribution $P_u^*$ can be calculated as following*

$$P_u^*(v) = P_u(v) \frac{\exp(g(u, v; \theta)/\alpha)}{\mathbb{E}_{w \sim P_u} [\exp(g(u, w; \theta)/\alpha)]} \tag{7}$$

The proof is placed in the appendix A.2. This lemma provides a close-formed expression of the most difficult distribution, greatly simplifying the implementation of DR-GNN. Specifically, modifications to the distribution of neighboring nodes can be simply realized

through the alteration of edge weights within the graph. Formally, the normalized adjacency matrix of the graph $\widetilde{A}$ is transformed into $\widetilde{A}'$ and

$$\widetilde{A}'_{ij} = \frac{\exp\left(g(i,j;\theta)/\alpha\right)}{\sum_{k \in \mathcal{N}(i)} \exp\left(g(i,k;\theta)/\alpha\right)} \widetilde{A}_{ij} \qquad (8)$$

The adjusted normalized adjacency matrix $\widetilde{A}'$, characterized by modified weights, is subsequently utilized to execute the aggregation operation pertaining to the graph. Given the equivalence between the GNN and the regularizer, this can be integrated during graph aggregation, necessitating only minor adjustments to the edge weights. In practice, we may suffer numerical instability due to the introduce of $\exp(.)$ in weights. To counteract this, we suggest to introduce the embedding normalization via $L2$ norm in calculating the weights, empirically yielding more stable results.

## 3.2 Graph Edge-Addition Strategy

Applying DRO in GNN-based recommendation incurs another challenge: Particularly, in DRO, all potential distributions $P$ should share the same support as the original distribution $P_u$, otherwise the KL-divergence would become infinite. This inherently implies that the support of $P$ would be restricted to the neighboring nodes, excluding the vast pool of non-neighboring nodes. The situation exacerbates given the typically sparse nature of recommendation data — most users may only have interactions with a handful of items. This significantly limits the flexibility and scope of potential distributions, increasing the risk of missing testing distribution and thereby undermining model robustness.

To address aforementioned issue, we propose a strategy called Graph Edge-Addition(GEA) that introduces slight perturbations $P_u^{add}$ in $P_u$ to expand its support. $P_u^{add}$ is defined over the support of all item set such that those non-neighboring nodes can be utilized for training better and robust embeddings. Formally, The objective of DR-GNN is improved as:

$$\mathcal{L}_{DRO\_smooth}(u) = \max_P \min_{P_u^{new}} \mathbb{E}_{v \sim P}\left[d_u g(u, v; \theta)\right]$$
$$\text{s.t.} \quad D_{KL}(P, P_u^{new}) \leq \eta \qquad (9)$$

where the new distribution $P_u^{new}$ is defined as $P_u^{new} = \gamma P_u + (1 - \gamma)P_u^{add}$, with $\gamma$ serving as a hyperparameter that controls the magnitude of the perturbations. Remarkably, we adjust $P_u$ towards minimization of regularizer rather than maximization. This is premised on the belief that a non-neighboring item, which exhibits greater similarity to user $u$, is more likely to be favored by the user. Such items are potentially useful in refining user embeddings.

The introduce of $p_u^{add}$ mitigates the inherent shortcomings of DRO in sparse recommendation datasets. It extends the range of potential distributions and harnesses the abundant information from non-neighboring nodes. Practically, the minimization optimization on $P_u^{new}$ can be formulated into finding the items minimizing $g(u, v; \theta)$. However, it can be computationally expensive, as it requires traversing over all nodes. To mitigate computational complexity, we employ a strategy of random sampling. Specifically, we select a subset of nodes randomly to form a candidate set, and then confine the traversal operation solely to this subset.

## 3.3 Theoretical Analyses

In this subsection, we provide a theoretical analysis to demonstrate the robustness of DR-GNN to distribution shift. For any user $u$, let $P_u^{ideal}$ denotes $u$'s ideal neighbor distribution used for model testing and the corresponding smoothness regularizer for node $u$ can be written as $\mathcal{L}_{ideal}(u; \theta) = \mathbb{E}_{v \sim P_u^{ideal}}\left[d_u g(u, v; \theta)\right]$.

THEOREM 3.1. Let $\widetilde{\mathcal{L}}_{DRO\_smooth}(u; \theta)$ serve as the estimation for $\mathcal{L}_{DRO\_smooth}(u; \theta)$. If $D_{KL}(P_u^{ideal}, P_u) \leq \eta$, then we have that with probability at least $1 - \delta$:

$$\mathcal{L}_{ideal}(u; \theta) \leq \widetilde{\mathcal{L}}_{DRO\_smooth}(u; \theta) + \mathcal{B}(q, d_u, \delta) \qquad (10)$$

where $\mathcal{B}(q, d_u, \delta) = \sqrt{\frac{8q \log\left(\frac{2ed_u}{q}\right) + 8 \log \frac{4}{\delta}}{d_u}}$ and $q$ is the Vapnik Chervonenkis dimension of the hypothesis space of parameter $\theta$.

The proof is presented in appendix A.3. Theorem 3.1 exhibits that the ideal loss is upper bound by empirical DRO loss if a sufficiently large data size is employed. At this point, we provide the theoretical guarantee for the resistance to distribution shift capability of DR-GNN.

## 3.4 Discussions

**The connection with APDA[52].** Interestingly, the edge weights introduced in our DR-GNN is highly similar with APDA. The key distinction lies in our choice to use initial embeddings for weight calculation, while APDA uses embeddings with multi-layer aggregation. This seemingly minor difference leads to a pronounced performance disparity in favor of DR-GNN over APDA in our experiments. The reason behind this is the theoretical grounding of our approach. Specifically, DR-GNN is derived from the theoretical-sound DRO framework, while APDA's design is predominantly heuristic and lacks a solid theoretical base.

Furthermore, our framework offers theoretical insights into several heuristic settings found in APDA:

Our framework also gives a theoretical explanations of many heretical settings used in APDA: 1) Operations such as the exponential function and symmetric normalization, which are adopted heuristically in APDA, can be indeed derived to the DRO objective. 2) APDA also heuristically employs a hyperparamter $\alpha$ to modulate the magnitude of the value in $\exp()$ [1]. In the context of DR-GNN, this $\alpha$ finds its theoretical counterpart — $\alpha$ serves as a Lagrange multiplier, acting as a surrogate hyperparameter to control the robust radius. We will discuss in depth the role of $\alpha$ and its relationship with the degree of distribution shift in section4.3.

**The connection with Attention Mechanism[32].** The attention mechanism has been adopted by recent work like Graph Attention Network (GAT) [33] to determine the edge weights. These weights in GAT, similar to ours, are predicated on node embeddings. However, there's a stark difference: while GAT employs a learnable non-linear layer to ascertain the weights, the weights in our DR-GNN are derived directly from DRO.

Despite the success of the attention mechanism across various domains, it underperforms in GNN-based recommendation. The primary reason lies in the sparsity and the lack of rich features of

---

[1]It's worth noting that this was not explicitly mentioned in their paper but was clearly implemented in their codes.

recommendation data. It heavily hinders the effective training of the attention function. This limitation is evident in our experimental results: GAT demonstrates suboptimal performance, while our DR-GNN exhibits effectiveness.

**The connection with GraphDA[8]** GraphDA is a method that enhances the adjacency matrix of a graph. This method initially pretrains a graph encoder to obtain the user/item embeddings following several iterations of graph convolution. Subsequently, the graph is reconstructed based on the similarity between these embeddings. This process facilitates denoising for active users and augmentating for inactive users. With the enhanced graph adjacency matrix, GraphDA retrains a randomly initialized graph encoder. Contrasting with GraphDA, the edge-adding operation in DR-GNN also reconstructs a new adjacency matrix, but with different motivations. The primary objective of GraphDA's graph enhancement is to equalize the number of neighbors for each node. This is achieved by denoising users with an excess of neighbors in the original data and augmenting users with a paucity of neighbors. Furthermore, GraphDA preemptively introduces edges between user-to-user and item-to-item to enable long-distance message passing. However, the integration of the GAE module aims to broaden the support of neighbor distribution. This expansion subsequently expand the uncertainty set of DRO, thereby endowing the model with enhanced generalization capabilities.

## 4 EXPERIMENTS

We aim to answer the following research questions:

- **RQ1:** How does DR-GNN perform compared with existing methods under various distribution shifts?
- **RQ2:** What are the impacts of the components (e.g., DRO on neighbour nodes, GEA) on DR-GNN?
- **RQ3:** How does the parameter $\alpha$ impacts DR-GNN?

**Datasets.** The experiments are conducted under three prevalent distribution shift scenarios: popularity shift, temporal shift, and exposure shift. Thus, eight datasets are employed for testing, namely Gowalla, Douban, AmazonBook, Yelp2018, Movielens-1M, Food, Coat, and Yahoo. For the popularity shift setting, we re-divide the train and test set of the dataset based on item popularity. The test set was designed in such a way that the popularity of all items approximated a uniform distribution, while a long-tail distribution was preserved within the training set. For the datasets under the temporal shift setting, we took the most recent 20% of interaction data from each user as the test set, and the earliest 60% of interaction data as the training set. Table 5 in appendix A.3 shows the statistics of each processed dataset.

**Baselines.** We use the conventional LightGCN as backbone and BPR loss for all the baselines. The methods compared in the study fall into several categories:

- **Methods against distribution shifts in Recommendation System(InvCF[47], BOD[38])** InvCF is the SOTA method on addressing the popularity shift through invariant learning. BOD achieves data denoising through bi-level optimization. Meanwhile, we acknowledge that there are some other methods to address the OOD problem, including CausPref[14], COR[36], HIRL[50], and InvPref[39]. However, these methods require additional information that is not available in our dataset and hence

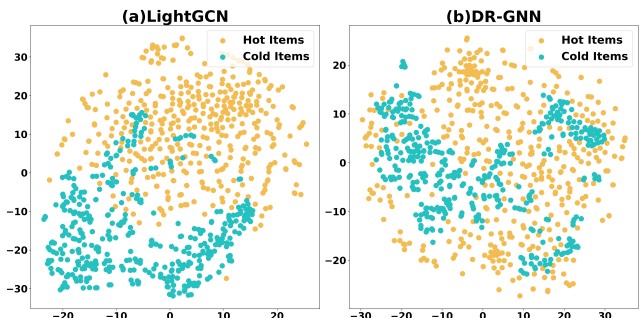

**Figure 3: t-SNE Visualization on Douban. DR-GNN ensures that the representations of hot items and cold items are almost distributed in the same space.**

cannot be tested. Certain methods among these necessitate pre-partitioned environmental datasets[50], others demand prior semantic information about users and items[14, 36], and some necessitate the assignment of environmental variables for each interaction, thereby rendering the strategy of random negative sampling inapplicable[39].

- **Graph contrastive learning methods(LightGCL[3], SGL[41])**. The methods use the contrastive learning on the graph and has been proven to alleviate the prevalent popularity bias.
- **Reconstructing the adjacency matrix methods(APDA[52], GAT[33], GraphDA[8])** Such methods reconstruct the adjacency matrix of the graph by adjusting edge weights or reconstructing the edges between nodes according to certain rules. Moreover, APDA and GAT can experience memory overflow issues on datasets with a large number of interactions. This is because GAT needs to calculate weights for each edge and perform backpropagation, while APDA needs to calculate weights for each layer of aggregation operations.

We used the source code provided in the original papers and searched for optimal hyperparameters for all comparison methods according to the instructions in the original papers.

**Evaluation Metrics.** Three commonly used metrics—Precision@$K$, Recall@$K$, and Normalized Discounted Cumulative Gain(NDCG@$K$) — are used to assess the quality of the recommendations, with $K$ being set by default at 20.

Further experimental details are presented in the appendixA.5.

### 4.1 Performance Comparison (RQ1 and RQ2)

In this section, we analyze the superior of DR-GNN under different distribution shift setting as compared with other baselines.

*4.1.1 Evaluations on Popularity Shift Setting.* Table 1 reports the comparison of performance on all the baselines under popularity shift. The majority of comparative methodologies fail to consistently yield satisfactory results across a variety of datasets, including algorithms specifically designed for popularity shift, such as APDA and InvCF. The improvement of APDA compared to Light-GCN is quite limited, indicating that its heuristic dynamic edge weight adjustment algorithm cannot handle popularity shift well.

**Table 1: The performance comparison on popularity shift datasets using LightGCN backbone. The best result is bolded and the runner-up is underlined. OOM stands for out of memory.**

| | Gowalla | | | Douban | | | Amazon-Book | | | Yelp2018 | | |
|---|---|---|---|---|---|---|---|---|---|---|---|---|
| | NDCG | Precision | Recall | NDCG | Precision | Recall | NDCG | Precision | Recall | NDCG | Precision | Recall |
| LightGCN(SIGIR20) | 0.0369 | 0.0170 | 0.0563 | 0.0792 | 0.0510 | 0.0723 | 0.0227 | 0.0129 | 0.0278 | 0.0136 | 0.0060 | 0.0221 |
| LightGCL(ICLR23) | 0.0380 | 0.0177 | 0.0593 | 0.0746 | 0.0464 | 0.0721 | 0.0243 | 0.0137 | 0.0305 | 0.0136 | 0.0061 | 0.0227 |
| SGL(SIGIR21) | 0.0381 | 0.0173 | 0.0595 | 0.0773 | 0.0493 | 0.0716 | 0.0232 | 0.0131 | 0.0296 | 0.0143 | 0.0063 | 0.0242 |
| InvCF(WWW23) | 0.0399 | 0.0181 | 0.0636 | 0.0740 | 0.0459 | 0.0725 | 0.0210 | 0.0118 | 0.0268 | 0.0144 | 0.0062 | 0.0243 |
| BOD(KDD23) | 0.0361 | 0.0169 | 0.0556 | 0.0815 | 0.0559 | 0.0686 | 0.0253 | 0.0141 | 0.0315 | 0.0145 | 0.0061 | 0.0253 |
| APDA(SIGIR23) | 0.0367 | 0.0168 | 0.0579 | OOM | OOM | OOM | 0.0229 | 0.0133 | 0.0286 | 0.0141 | 0.0062 | 0.0228 |
| GAT(ICLR18) | 0.0227 | 0.0104 | 0.0334 | OOM | OOM | OOM | OOM | OOM | OOM | OOM | OOM | OOM |
| GraphDA(SIGIR23) | 0.0341 | 0.0166 | 0.0570 | 0.0892 | 0.0596 | 0.0808 | 0.0199 | 0.0113 | 0.0262 | 0.0140 | 0.0059 | 0.0243 |
| DR-GNN | **0.0517** | **0.0238** | **0.0774** | **0.1044** | **0.0704** | **0.0905** | **0.0327** | **0.0183** | **0.0402** | **0.0193** | **0.0086** | **0.0322** |

**Table 2: The performance comparison on temporal shift datasets using LightGCN backbone. The best result is bolded and the runner-up is underlined.**

| | MovieLens-1M | | | Food | | |
|---|---|---|---|---|---|---|
| | NDCG | Precision | Recall | NDCG | Precision | Recall |
| LightGCN | 0.2041 | 0.1741 | 0.1310 | 0.0429 | 0.0168 | 0.0585 |
| LightGCL | 0.2055 | 0.1750 | 0.1335 | 0.0421 | 0.0165 | 0.0574 |
| SGL | 0.2177 | 0.1830 | 0.1407 | 0.0401 | 0.0159 | 0.0553 |
| InvCF | 0.2138 | 0.1759 | 0.1354 | 0.0411 | 0.0161 | 0.0557 |
| BOD | 0.2116 | 0.1729 | 0.1343 | 0.0436 | 0.0170 | 0.0592 |
| APDA | **0.2362** | **0.1954** | **0.1518** | 0.0400 | 0.0161 | 0.0554 |
| GAT | 0.1942 | 0.1713 | 0.1325 | 0.0316 | 0.0131 | 0.0464 |
| GraphDA | 0.2213 | 0.1866 | 0.1482 | 0.0411 | 0.0161 | 0.0580 |
| DR-GNN | 0.2330 | 0.1924 | 0.1452 | **0.0440** | **0.0172** | **0.0592** |

**Table 3: The performance comparison on exposure shift datasets using LightGCN backbone. The best result is bolded and the runner-up is underlined.**

| | Coat | | | Yahoo | | |
|---|---|---|---|---|---|---|
| | NDCG | Precision | Recall | NDCG | Precision | Recall |
| LightGCN | 0.0802 | 0.0243 | 0.1576 | 0.0736 | 0.0118 | 0.1504 |
| LightGCL | 0.0839 | 0.0243 | 0.1577 | 0.0734 | 0.0118 | 0.1476 |
| SGL | 0.0839 | 0.0245 | 0.1631 | 0.0740 | 0.0122 | 0.1548 |
| InvCF | 0.0842 | 0.0241 | 0.1668 | 0.0742 | 0.0122 | 0.1539 |
| BOD | 0.0817 | 0.0219 | 0.1542 | **0.0816** | 0.0115 | 0.1481 |
| APDA | 0.0821 | 0.0243 | 0.1706 | 0.0756 | 0.0124 | 0.1588 |
| GAT | 0.0836 | 0.0243 | 0.1633 | 0.0722 | 0.0119 | 0.1500 |
| GraphDA | 0.0847 | 0.0245 | 0.1641 | 0.0748 | 0.0117 | 0.1528 |
| DR-GNN | **0.0874** | **0.0255** | **0.1744** | 0.0774 | **0.0126** | **0.1624** |

Meanwhile, GAT, which also dynamically adjusts edge weights, performs much worse than LightGCN, suggesting that the attention mechanism may inadvertently intensify the impact of distribution shift. Our proposed method, DR-GNN, consistently and significantly surpasses the state-of-the-art benchmarks across all popularity shift datasets. The robustness of DR-GNN is ascribed to the incorporation of uncertainty inherent in observed data distributions by DRO. This feature enables the model to perform optimally under various environments, rather than solely relying on training data.

**Visualization Results.** As shown in Figure 3, to better understand how DR-GNN handles distribution shift, we perform t-SNE visualization[31] of the item embeddings on the Douban dataset. According to the ranking results of item popularity, the top 5% most popular items are selected as hot items and the bottom 5% as cold items. It can be observed that in the representations learned by LightGCN, there is a clear gap between hot items and cold items, while both are distributed in the same space in the representations learned by DR-GNN. This suggests that DR-GNN eliminate the impact of distribution shift caused by popularity shift.

*4.1.2 Evaluations on Temporal Shift Setting.* Temporal bias takes into account changes in user interests over time. It is more complex than the distribution shift caused by popularity, as it encompasses factors beyond popularity alone. We simulate temporal bias by dividing training and test sets for each user according to the timestamp of the interaction. Dataset Food and Movielens-1M are used in this setting.

In the context of the temporal shift setting, we noticed that on the Movielens-1M dataset, although APDA performed better than DR-GNN, the improvement is not significant, and its performance was relatively unstable, evidenced by its inferior performance compared to LightGCN on the Food dataset. Similar observations were made for GraphDA. Compared to most other benchmark methods, DR-GNN continues to consistently deliver good performance.

*4.1.3 Evaluations on Exposure Shift Setting.* In practical contexts, users are typically exposed to a limited subset of items, thereby neglecting a significant majority. This phenomenon named "exposure bias" suggests that the absence of user interaction with specific items does not unequivocally signify their disinterest. Consequently, in real-world datasets, the patterns of missing user interaction records are not randomly distributed, but rather, they are "missing-not-at-random". Here we conduct experiments on widely used missing-complete-at-random datasets: Yahoo and Coat.

**Table 4: Ablation Study on Gowalla and Yelp2018**

| Dataset | Method | NDCG | Precision | Recall |
|---------|--------|------|-----------|--------|
| Gowalla | LightGCN | 0.0369 | 0.0170 | 0.0563 |
| | DR-GNN w/o DRO | 0.0391 | 0.0174 | 0.0583 |
| | DR-GNN w/o GEA | 0.0494 | 0.0231 | 0.0761 |
| | DR-GNN | **0.0517** | **0.0238** | **0.0774** |
| Yelp2018 | LightGCN | 0.0136 | 0.0060 | 0.0221 |
| | DR-GNN w/o DRO | 0.0161 | 0.0070 | 0.0265 |
| | DR-GNN w/o GEA | 0.0182 | 0.0081 | 0.0305 |
| | DR-GNN | **0.0193** | **0.0086** | **0.0322** |

According to table 3, DR-GNN can also handle distribution shift caused by exposure bias well. Besides, we noticed that on the Yahoo dataset, BOD's NDCG metric surpasses other comparison methods, but it is weaker than LightGCN in terms of Precision and Recall metrics. We believe this might be due to BOD's weight adjustment strategy overfitting to certain interactions, causing them to rank higher, which leads the recommendation model to concentrate too much on a few highly relevant results. Furthermore, BOD's performance on the coat dataset is only slightly better than LightGCN, indicating that its performance is not stable.

The experiments under the aforementioned three settings demonstrate that our method can handle different types of distribution shifts.

## 4.2 Ablation Study (RQ3)

We conducted an ablation study to investigate the effects of different modules in DR-GNN, including the DRO and GEA modules. We compared the DR-GNN with its two variants "DR-GNN w/o DRO" and "DR-GNN w/o GEA" based on whether the DRO and GEA modules were enabled. The results in Table 4 demonstrate that the use of DRO for graph aggregation operations can significantly enhance the model's performance, and GEA further improves the model's effectiveness. Surprisingly, we also found that the simple use of GEA could also yield some gains. This may be attributed to the additional edges introduced by GEA, which have accelerated the convergence of smoothness regularization. The ablation study highlights the fact that all modules in DR-GNN can boost the model's learning.

## 4.3 Role of the parameter $\alpha$ (RQ4)

The parameter $\alpha$ in Eq.(7) plays an important role in the DRO. Based on the previous derivation, the role of $\alpha$ is to control the size of uncertainty set. A smaller $\alpha$ indicates a larger uncertainty set for optimizing. However, an excessively large search space can easily lead to model overfitting. As $\alpha$ tends towards zero, DRO will excessively amplify the weight of the node with the maximum smooth regularization loss, which can easily lead to overfitting to the meaningless distribution. When $\alpha$ is infinitely large, the worst-case distribution tends towards a uniform distribution. At this point, DR-GNN without GEA is equivalent to LightGCN. Therefore, $\alpha$ is an important hyperparameter that needs to be carefully tuned. Different values

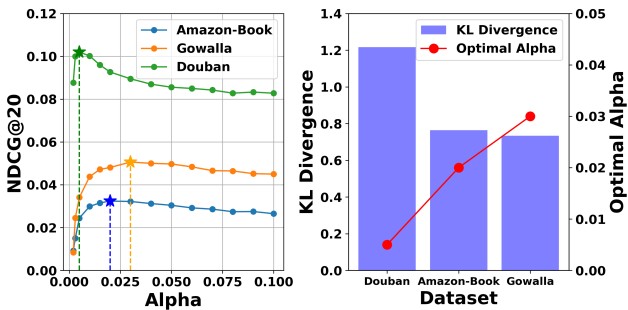

**Figure 4: Analysis of the role of $\alpha$. (a) Left: The performance of DR-GNN in terms of NDCG across different $\alpha$ on three datasets with varying degrees of distribution shift. (b) Right: The relationship between the degree of distribution shift and the optimal $\alpha$.**

need to be set depending on the degree of distribution shift in the dataset. Empirically, we search for $\alpha$ within the range of $[0, 1]$.

Besides, we empirically verified the relationship between the optimal $\alpha$ and the degree of distribution shift on the popularity shift dataset. We utilize the KL-divergence between item frequencies as a metric to measure the degree of distribution shift between the training and testing sets, and then tried to train the model by tuning different $\alpha$ parameters. Figure 4(a) shows the NDCG performance of DR-GNN with different $\alpha$ under varying degrees of popularity shift in different datasets. It can be observed that either excessively large or small $\alpha$ can impact the performance, and the optimal value varies across different datasets. In Figure 4(b), we further quantified the distribution shift using KL divergence. It was found that datasets with a larger degree of shift require a smaller optimal $\alpha$, as they necessitate a larger uncertainty set. This corresponds to our previous explanation.

## 5 RELATED WORK

### 5.1 GNN-based Recommender System

In recent years, graph-based recommendation systems have attracted extensive attention and research. Such systems leverage the structure of graphs to discover and infer user preferences and interests, thereby providing more personalized and accurate recommendations. Compared to the traditional collaborative filtering method that only use first order information of node, the GNN can capture higher order signal in the interactions by aggregating the information from neighboring nodes. LightGCN[10] simplify the original stucture of GCN[16] by dropping the feature transformation nonlinear activation and self-connection. NIA-GCN[29] improves the aggregation way in the GCN. They use PNA aggregator to model more complex interactions in the graph data.

Contrastive learning has also been extensively applied in graph-based recommendation models. SGL[41] applies data augmentation to graph-structured data through node dropout, edge dropout and random walk, and constructs the positive and negative pairs using the different views of the nodes' embedding. Other CL-based recommenders mainly improve the way of augmenting the graph.

LightGCL[3] uses SVD to generate new graph structures, which emphasizes the important signal in the user/item interactions. SimGCL[44] and XSimGCL[43] uses the random noise-based data augmentation on embeddings avoiding the popularity bias by making features more uniformly distributed.

Some methods are designed to target the OOD problem on GNN-based recommendation models. APDA[52] dynamically adjusts the edge weights of the graph, reducing the impact of popular items while amplifying the influence of unpopular items. GraphDA[8] reconstructs the adjacency matrix of the graph through a pre-trained encoder, thereby achieving signal augmentation and denoising within the graph. RGCF[30] improves graph structure learning by identifying more reliable message-passing interactions, while simultaneously maintaining the diversity of the enhanced data.

Despite these efforts, the aforementioned methods can only address specific types of OOD problems, that is, they only focus on the distribution shift resulting from certain specific factors, such as noise and popularity bias, hence are not generic. Moreover, these methods are mostly based on heuristic rule design and lack theoretical guarantees.

## 5.2 OOD in Recommender System

When training models, it is a common assumption that test data originates from the same distribution as the training data. However, in real-world scenarios, models may encounter test data that deviates from the distribution of the training data - a phenomenon known as Out-of-Distribution. The presence of OOD data can potentially precipitate a degradation in the model's performance. There are many methods used to solve the OOD problem in recommender systems.

One category of methodologies endeavors to alleviate the impact of distribution shift through the identification of invariant components within embeddings. InvCF[47] effectively mitigates the impact of popularity shifts by incorporating an auxiliary classifier that formulates recommendations predicated on popularity. InvPref[39] and HIRL[50] seperate the dataset into multiple environments by attributing an environment variable to each interaction and then leverage invariant learning to automatically identify elements that remain constant irrespective of the environment. Some of the works employ causal learning in addressing the OOD problem. COR[36] uses causal graph modeling and counterfactual reasoning to address the effect of out-of-date interactions. CausPref[14] utilizes a differentiable causal graph learning approach to obtain the invariant user preferences. Other works, such as BOD[38] applies bi-level optimization to ascertain the weight for each interaction to negate the effects of noisy interactions.

However, the kind of works are not specifically designed for GNN-based recommender system and thus fail to address the impact of distribution shifts on the structure of the graph models themselves.

## 5.3 Distributionally Robust Optimization

Distributionally Robust Optimization is a method for addressing OOD problems. The primary goal of DRO is to find a solution that is not only optimal for the given data distribution but also robust against variations in the data distribution, i.e. the family of distributions consisting of all distributions within a certain distance from the current data distribution. Many measures of distribution distance are used in DRO, including KL-divergence[15], Wasserstein-distance[26], MMD distance[28]. It has been found that DRO tends to induce model overfitting to the noisy samples[46] and GroupDRO[23] was introduced to address this issue.

DRO has also been applied in the field of RS. S-DRO[40] categorizes users into different groups based on the popularity of the items they interact with, and then employs group DRO to improve long-term fairness for disadvantaged subgroups. DROS[42] applies DRO to sequential recommendation tasks to address the OOD problem in the streaming of recommendation data. These methods are not used on GNN-based RS.

Some research has explored the application of DRO on GNN[6, 22, 48]. On one hand, these methods fundamentally aim to address distributional shifts caused by noise in node embeddings, which is distinct from our approach that considers distributional shifts within the graph's topological structure. On the other hand, these methods are not applied in recommendation systems and do not take into account the challenges posed by the unique characteristics of recommendation datasets when applying DRO.

## 6 CONCLUSION

This paper proposes an method DR-GNN, which introduces DRO into the aggregation operation of GNN, to solve the problem of existing graph-based recommendation systems being easily affected by distribution shift. Based on comparative experiments under multiple datasets and settings, as well as visualization studies on real datasets, DR-GNN has been proven to effectively solve the distribution shift problem, enhancing the robustness of graph models.

A direction worthy of exploration in future work is the application of DRO based on other distance metrics, such as the Wasserstein distance, MMD distance, etc., to address some of the shortcomings of the KL divergence-based DRO discussed in this paper. We believe that DR-GNN could provide a new perspective for future work aimed at enhancing graph-based RS robustness.

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

## A APPENDICES

### A.1 The proof of Lemma 1

PROOF. Eq(.3) can be written as follows,

$$\mathcal{L}_{smooth} = \frac{1}{2} \sum_u \sum_{v \in \mathcal{N}(u)} \left\| \frac{E_u}{\sqrt{d_u}} - \frac{E_v}{\sqrt{d_v}} \right\|_F^2 = \text{tr}\left(E^\top LE\right) \quad (11)$$

We first compute the derivative of the Eq(.11)

$$\frac{\partial \mathcal{L}}{\partial E} = 2cLE = 2cE - 2c\widetilde{A}E \quad (12)$$

In order to minimize Eq(.11) we take multi-step gradient descent. The $k$-th step is as follows:

$$E^{(k)} \leftarrow E^{(k-1)} - b \cdot \left.\frac{\partial \mathcal{L}}{\partial E}\right|_{E=E^{(k-1)}}$$

$$= (1 - 2bc)E^{(k-1)} + 2bc\widetilde{A}E^{(k-1)} \quad (13)$$

When we set the learning rate $b$ as $\frac{1}{2c}$, we have the following iterative steps:

$$E^{(k)} \leftarrow \widetilde{A}E^{(k-1)} \quad (14)$$

which is equivalent to one-step aggregation operation in LightGCN. Thus the multilayer aggregation operation is equivalent to minimizing the smoothness regularization with a multi-step gradient descent algorithm. □

### A.2 The proof of Lemma 2

In this section we show the derivation of a multi-layer optimization problem in DRO converted to a single-layer optimization problem. The original DRO question is as follows,

$$\min_\theta \mathcal{L}_{DRO\_smooth}(u) = \min_\theta \max_P \mathbb{E}_{v \sim P}\left[d_u g(u, v; \theta)\right]$$
$$\text{s.t. } D_{KL}(P, P_u) \leq \eta \quad (15)$$

Given that the presence of $d_u$ as a constant does not affect optimization, we will omit it in our discussion, leading to the following form of optimization problem,

$$\max_P \mathbb{E}_{v \sim P}\left[g(u, v; \theta)\right]$$
$$\text{s.t. } D_{KL}(P, P_u) \leq \eta \quad (16)$$

We focus on how to eliminate the inner maximisation optimisation problem and the distributional constraint term. Assume $L(j) = Q(j)/P_u(j)$ and define a convex function $\phi(x) = x \log x - x + 1$. Then the divergence $D_{KL}(Q, P_u)$ can be written as $\mathbb{E}_{P_u}[\phi(L)]$. The inner layer maximization optimization problem can be reformulated as follow:

$$\max_L \mathbb{E}_{v \sim P_u}\left[g(u, v; \theta)L\right]$$
$$\text{s.t. } \mathbb{E}_{P_u}[\phi(L)] \leq \eta, \mathbb{E}_{P_u}[L] = 1 \quad (17)$$

As a convex optimization problem, we use the Lagrangian function to solve it:

$$\min_{\alpha \geq 0, \beta} \max_L \mathbb{E}_{v \sim P_u}\left[g(u, v; \theta)L\right] - \alpha(\mathbb{E}_{P_u}[\phi(L)] - \eta) + \beta(\mathbb{E}_{P_u}[L] - 1)$$

$$= \min_{\alpha \geq 0, \beta} \left\{ \alpha\eta - \beta + \alpha \max_L \mathbb{E}_{v \sim P_u}\left[\frac{g(u, v; \theta) + \beta}{\alpha}L - \phi(L)\right] \right\}$$

$$= \min_{\alpha \geq 0, \beta} \left\{ \alpha\eta - \beta + \alpha \mathbb{E}_{v \sim P_u}\left[\max_L \left(\frac{g(u, v; \theta) + \beta}{\alpha}L - \phi(L)\right)\right] \right\}$$

Notice that $\max_L \left(\frac{g(u,v;\theta)+\beta}{\alpha}L - \phi(L)\right) = \phi^*(\frac{g(u,v;\theta)+\beta}{\alpha})$ is the convex conjugate function of $\phi(x)$ and we have $\phi^*(x) = e^x - 1$. $L(v) = e^{\frac{g(u,v;\theta)+\beta}{\alpha}}$ when the maximum value is obtained.

$$\min_{\alpha \geq 0, \beta} \left\{ \alpha\eta - \beta + \alpha \mathbb{E}_{v \sim P_u}\left[\max_L \left(\frac{g(u, v; \theta) + \beta}{\alpha}L - \phi(L)\right)\right] \right\}$$

$$= \min_{\alpha \geq 0, \beta} \left\{ \alpha\eta - \beta + \alpha \mathbb{E}_{v \sim P_u}\left[ e^{\frac{g(u,v;\theta)+\beta}{\alpha}} - 1 \right] \right\}$$

$$= \min_{\alpha \geq 0} \left\{ \alpha\eta + \alpha \log \mathbb{E}_{v \sim P_u}\left[ e^{\frac{g(u,v;\theta)}{\alpha}} \right] \right\}$$

where $\beta = -\alpha \log \mathbb{E}_{v \sim P_u}\left[ e^{\frac{g(u,v;\theta)}{\alpha}} \right]$ and $L(v) = \frac{e^{\frac{g(u,v;\theta)}{\alpha}}}{\mathbb{E}_{w \sim P_u}\left[ e^{\frac{g(u,w;\theta)}{\alpha}} \right]}$

when the minimum value is obtained. We consider the Lagrange multiplier $\alpha$ as a hyperparameter related to the robustness radius to obtain the final unconstrained single-layer optimization problem.

$$\min_\theta \mathcal{L}_{DRO\_smooth}(u) = \min_\theta \left\{ \alpha\eta + \alpha \log \mathbb{E}_{v \sim P_u} \exp\left(\frac{g(u, v; \theta)}{\alpha}\right) \right\} \quad (18)$$

where the worst-case distribution

$$P_u^*(v) = P_u(v) \frac{\exp\left(g(u, v; \theta)/\alpha\right)}{\mathbb{E}_{w \sim P_u}\left[\exp(g(u, w; \theta)/\alpha)\right]}$$

.

### A.3 The proof of Theorem 3.1

PROOF.

$$\mathcal{L}_{ideal}(u; \theta) = \mathbb{E}_{v \sim P_u^{ideal}}\left[d_u g(u, v; \theta)\right]$$
$$\leq \mathcal{L}_{DRO\_smooth}(u; \theta) \quad (19)$$
$$\leq \widetilde{\mathcal{L}}_{DRO\_smooth}(u; \theta) + \mathcal{B}(q, d_u, \delta)$$

It is obvious that the first inequality holds, because as long as $P_u^{ideal}$ exists in the uncertainty set, the $\mathcal{L}_{DRO\_smooth}(u; \theta)$ must be greater than the ideal loss $\mathcal{L}_{ideal}(u; \theta)$. The second inequality follows from the relationship between empirical error and expected error[2]. □

### A.4 Dataset statistics

The statistics of each dataset used in the experiment are shown in Table 5, which lists the number of users, items, interactions, and sparsity of the dataset. The brief introductions of all datasets are as follows:

- **Gowalla[11]**. Gowalla is the check-in dataset obtained from Gowalla[2].
- **Douban[27]**. This dataset is collected from a popular review website Douban[3] in China. We transform explicit data into implicit using the same method as applied in Movielens.
- **Amazon-Book[37]**. The Amazon-Book dataset is a comprehensive collection of data that primarily focuses on book reviews from the Amazon platform[4]. This dataset is often used in research, particularly in the field of recommendation systems.
- **Yelp2018[10]**. Yelp2018[5] is from the 2018 edition of the Yelp challenge, containing Yelp's bussiness reviews and user data.

---

[2]https://www.gowalla.com/
[3]https://www.douban.com/
[4]https://www.amazon.com/
[5]https://www.yelp.com/dataset

**Table 5: Dataset statistics.**

| Setting | Dataset | #Users | #Items | #Interactions | Density |
|---|---|---|---|---|---|
| Popularity Shift | Gowalla | 29858 | 40705 | 1024507 | 0.0008 |
| | Douban | 61788 | 7768 | 10449897 | 0.0218 |
| | Amazon-Book | 52643 | 89807 | 2964807 | 0.0006 |
| | Yelp2018 | 77277 | 41978 | 2062568 | 0.0006 |
| Temproal Shift | Movielens-1M | 6040 | 3653 | 880999 | 0.0399 |
| | Food | 7450 | 10977 | 251038 | 0.0031 |
| Exposure Shift | Coat | 290 | 284 | 2745 | 0.0333 |
| | Yahoo | 14382 | 1000 | 129748 | 0.0090 |

- **Movielens-1M[45]**. Movielens is the widely used dataset from[] and is collected from MovieLens[6]. We use the version of 1M. We transform explicit data to implicit feedback by treating all user-item ratings as positive interactions.
- **Food[51]**. Food datasets contain recipe details and reviews from Food.com(formerly GeniusKitchen)[7]. Data includes cooking recipes and review texts.
- **Coat[19] & Yahoo[24]**. These two datasets are obtained from the Yahoo music and Coat shopping recommendation service, respectively. Both datasets contain a training set of biased rating data collecting from the normal user interactions and a test set of unbiased rating data containing user ratings on randomly selected items. The rating data are translated to implicit feedback, *i.e.,* , the rating larger than 3 is regarded as positive.

## A.5 Parameter Settings.

For a fair comparison, the embedding size is fixed to 32 and layer num is set as 3 for all comparison methods. TPE-based Bayesian optimiser is used to search for the optimal hyperparameters. Specifically, as for DR-GNN, the learning rate is tuned from $\{0.1, 0.01\}$ and the coefficient of $L_2$ regularization term $\lambda$ is tuned from $\{0, 0.1, 0.01, ..., 1e-6\}$. The $\tau$ in BPR loss is searched from $\{0.1, 0.2, ..., 1.0\}$. The $\alpha$ in DRO is search from $(0, 1)$. The coefficient $\gamma$ in GEA is search from $\{0.1, 0.2, 0.3, 0.4, 0.5\}$.

We utilize the all-ranking strategy, in which all items, excluding the positive ones from the training set, are ranked by the recommendation model for each user.

Received 20 February 2007; revised 12 March 2009; accepted 5 June 2009

---

[6]https://movielens.org/
[7]https://cseweb.ucsd.edu/ jmcauley/datasets.html#foodcom

