# OpenReview forum: "Distributionally Robust Graph-based Recommendation System"
_ACM.org/TheWebConf/2024/Conference — TheWebConf24 Oral_

### Official Review · Reviewer_SzPa · 2023-11-19

**Novelty:** 6
**Technical Quality:** 6

**Review:**

**Summary**

This paper addresses the limitations of GNNs in Recommender Systems, particularly their reliance on the assumption that training and testing data share the same distribution. The authors propose a novel method named DR-GNN, which integrates Distributional Robust Optimization into GNN-based recommendation. DR-GNN is designed to enhance the robustness of GNNs against distribution shifts that are common in RS due to dynamic user preferences and biases in data collection.

**Strengths**

1. The paper addresses a significant and underexplored issue in the field of recommender systems - the impact of distribution shifts on GNN-based methods. This is a timely and relevant topic given the dynamic nature of user preferences in real-world applications.

2. The approach in reinterpreting GNN as a graph smoothing regularizer to facilitate the application of DRO is a smart move. This not only addresses a core challenge but also enhances the robustness of GNNs against shifts in neighbor distributions.

3. The paper is strong in both theoretical analyses and empirical experiments. The experiments validate the effectiveness of DR-GNN against typical distribution shifts, providing a solid foundation for their claims.

**Weaknesses**

1. Some sections of the paper, especially the introduction and methodological explanations, could benefit from clearer and more concise language.
2. While the explanation of DR-GNN is detailed, it may benefit from further simplification or a more step-by-step breakdown, especially in describing how DRO is integrated into the GNN framework.
3. The paper mentions potential numerical instability due to the introduction of exponential weights. It might be insightful to elaborate on how these challenges are addressed and whether they impact the model’s performance or scalability.

**Questions:**

see weakness.

**Reviewer Confidence:**

2: The reviewer is willing to defend the evaluation, but it is likely that the reviewer did not understand parts of the paper

**Scope:**

4: The work is relevant to the Web and to the track, and is of broad interest to the community

---

### Official Review · Reviewer_oZJ3 · 2023-11-22

**Novelty:** 4
**Technical Quality:** 4

**Review:**

In this paper, the authors propose DR-GNN that integrates Distributional Robust Optimization (DRO) with GNN for recommendations. DR-GNN tackles two main challenges: (1) It reinterprets GNNs as graph smoothing regularizers, which aligns them with the principles of DRO for better applicability to graph data; and (2) It addresses the issue of sparse recommendation data, which can hinder robust optimization, by introducing small perturbations to the training distribution to broaden its support. Despite the intricate optimization involved, DR-GNN is designed for easy and efficient implementation. The comprehensive experiments demonstrate DR-GNN's superior performance in dealing with various types of distribution shifts.

**Questions:**

1. The presentation of this paper requires significant improvement. There are numerous grammatical errors and typographical mistakes present throughout. For example, "in field of recommendation systems"->"in the field of recommendation systems"; "... in collaborative recommendation However, a pervasive assumption..."->" ...in collaborative recommendation. However, a pervasive assumption..."; "in practical"->"in practice"; "Movielens is the widely used dataset from[]" misses the reference.

2. The authors have not made the code available nor provided pseudocode, which raises concerns regarding the reproducibility of their work.

3. The paper lacks some essential sensitivity analysis experiments. For example, there is no sensitivity analysis for the parameter $\gamma $ in GEA, embedding size, the number of GNN layers, and coefficient $\lambda$.

4. The distinction between "distribution shift" and "exposure bias" has been elucidated in the paper.

**Ethics Review Description:**

No.

**Reviewer Confidence:**

3: The reviewer is confident but not certain that the evaluation is correct

**Scope:**

3: The work is somewhat relevant to the Web and to the track, and is of narrow interest to a sub-community

---

### Official Review · Reviewer_sNmo · 2023-11-23

**Novelty:** 4
**Technical Quality:** 6

**Review:**

Effectively points out the problems associated with frequent distribution shifts in real-world scenarios and provides a clear depiction of strategies to rationally address these issues. Particularly noteworthy is the thorough analysis of the problem targeted in this study using strategic numerical analyses, allowing for a proper understanding of the necessity of the research. Additionally, the paper critically examines the limitations of existing research and explicitly mentions the necessary aspects to overcome these limitations, providing a clear understanding of the proposed approach.
The incorporation of theoretical insights into the fusion of DRO and GNN contributes to the robustness of the paper, creating a sense of depth in the presented research.
Overall, the logical flow of the paper is clean, and there were no significant difficulties in comprehending the content. However, the ablation study conducted on a relatively limited dataset raises concerns about the feasibility of assessing the impact of each module.

**Questions:**

I'm curious why the Beauty benchmark dataset, a well-known dataset commonly used in similar studies, was not employed in conducting this research.

**Reviewer Confidence:**

3: The reviewer is confident but not certain that the evaluation is correct

**Scope:**

2: The connection to the Web is incidental, e.g., use of Web data or API

---

### Official Review · Reviewer_wf42 · 2023-11-26

**Novelty:** 6
**Technical Quality:** 6

**Review:**

This paper introduces DR-GNN, aiming to enhance the robustness of LightGCN with distributionally robust optimization.
Specifically, the author firstly proves that performing graph aggregation in LightGCN is equivalent to optimizing a graph smoothness regularizer, and adopts DRO on this smoothness regularizer loss to enhance the robustness of LightGCN. After that, the author adopts GEA to further solve the problem of data sparsity.
Generally, this paper is novel and well organized. But there are some problems in this paper, needing further discussion. My concerns are about the reproducibility and some experiment details. If the author can provide the code and solve my questions, I will raise my score for Technical Quality.
Strength:
1.	The motivation in this paper is intuitive and reasonable.
The author firstly studies the problem of GNN from an aspect of graph filtering, and then points out the problems from both out-of-distribution problem and data sparsity to introduce proposed edge reweighting and GEA.
2.	Although the theories are complicated, the implementation is quite easy.
3.	The experiments in this paper are sufficient to support the effectiveness of the proposed idea.
Three commonly used data settings and latest baselines are included, and the performance of DR-GNN is stable and excellent.
4.	I especially appreciate the discussion section.
This paper links the proposed DR-GNN with previous studies, which is convincing and inspiring. The advantages and distinction of DR-GNN are elucidated clearly.
5.	The study of the relationship between  $\alpha$ and the KL divergence is interesting and convincing.


Weakness:
1.	Some experiment details and necessary related works are missing.
For example, the uniform distribution mentioned in line 505 lacks a detailed description. Also, necessary citations for this widely adopted uniform data preprocessing are missing [1][2][3].
Moreover, I kindly suggest the author add discussions from the perspective of recommendation debias in the Related Work section.
2.	The code is not provided.
3.	There are some format problems in this paper, please kindly correct them.
For example, a blank space is missing between the citation and the last word. (i.e., LightGCN [10] rather than LightGCN[10] in line 801)
[1] Tianxin Wei, Fuli Feng, Jiawei Chen, Ziwei Wu, Jinfeng Yi, and Xiangnan He. Model-agnostic
counterfactual reasoning for eliminating popularity bias in recommender system. In KDD, 2021.
[2] Yu Zheng, Chen Gao, Xiang Li, Xiangnan He, Yong Li, and Depeng Jin. Disentangling user interest and conformity for recommendation with causal embedding. In WWW, 2021.
[3] Zhang A, Ma W, Wang X, et al. Incorporating bias-aware margins into contrastive loss for collaborative filtering[J]. Advances in Neural Information Processing Systems, 2022, 35: 7866-7878.

**Questions:**

1. Is this Lemma 1 specially tailored for LightGCN? If so, can the theories be easily adapted to other graph based backbones like NGCF [1] and UltraGCN [2]? If the theories can be easily adapted, can the implementation of edge reweighting and GEA also gain the effectiveness over these backbones?
2.	Since the proposed method is loss-agnostic, can the proposed method also gain over other loss functions like SSM loss [3]?
3.	Does the random sampling mentioned in line 403 mean sample some nodes from whole nodes? Can you give more details about this? For example, what is the ratio of sampling.
4.	Can you provide an intuitive example (e.g., a figure) about adding new edges into the original graph? Does it mean we traverse over the whole candidate set to solve the problem defined in equation (9) to get the best new edges? If so, what is the computational cost of this operation? I believe that this operation highly depends on the scale of the candidate set and it may also be time consuming. Can you give the time consumption of the proposed method, compared to other baselines? Either time cost for each training epoch or the time complexity analysis is fine.
5.	Have you ever studied the effect of $\gamma$ for hyperparameter study?
6.	I’m just also curious about if this method is specially tailored for recommendation? Because I think the proofs in this paper are quite general and may also be easily adapted to other tasks.
7.	Can you provide more details about the details of experiment settings, such as gpu devices and batch size? I notice that both GAT and APDA encounter the out-of-memory problem. What is the reason for this? I don’t think these models are too large to implement, by reducing the batch size.

[1] Wang X, He X, Wang M, et al. Neural graph collaborative filtering[C]//Proceedings of the 42nd international ACM SIGIR conference on Research and development in Information Retrieval. 2019: 165-174.
[2] Mao K, Zhu J, Xiao X, et al. UltraGCN: ultra simplification of graph convolutional networks for recommendation[C]//Proceedings of the 30th ACM International Conference on Information & Knowledge Management. 2021: 1253-1262.
[3] Wu J, Wang X, Gao X, et al. On the effectiveness of sampled softmax loss for item recommendation[J]. arXiv preprint arXiv:2201.02327, 2022.

**Reviewer Confidence:**

3: The reviewer is confident but not certain that the evaluation is correct

**Scope:**

4: The work is relevant to the Web and to the track, and is of broad interest to the community

---

### Decision · Program_Chairs · 2024-01-22

**Decision:**

Accept (Oral)

**Comment:**

This paper presents DR-GNN, which seeks to improve the resilience of LightGCN through distributionally robust optimization. The reviewers generally concur that the work makes a substantial and well-articulated contribution. It is advised that the authors incorporate the contents of their rebuttal into the paper to further solidify the work.